# The Effects of Chimeric Antigen Receptor (CAR) Hinge Domain Post-Translational Modifications on CAR-T Cell Activity

**DOI:** 10.3390/ijms23074056

**Published:** 2022-04-06

**Authors:** Sachiko Hirobe, Keisuke Imaeda, Masashi Tachibana, Naoki Okada

**Affiliations:** 1Laboratory of Clinical Pharmacology and Therapeutics, Graduate School of Pharmaceutical Sciences, Osaka University, 1-6 Yamadaoka, Suita 565-0871, Osaka, Japan; hirobe-s@phs.osaka-u.ac.jp; 2Department of Molecular Pharmaceutical Science, Graduate School of Medicine, Osaka University, 2-2 Yamadaoka, Suita 565-0871, Osaka, Japan; 3Department of Pharmacy, Osaka University Hospital, 2-15 Yamadaoka, Suita 565-0871, Osaka, Japan; 4Project for Vaccine and Immune Regulation, Graduate School of Pharmaceutical Sciences, Osaka University, 1-6 Yamadaoka, Suita 565-0871, Osaka, Japan; imakei0716@gmail.com (K.I.); tacci@phs.osaka-u.ac.jp (M.T.); 5Laboratory of Vaccine and Immune Regulation (BIKEN), Graduate School of Pharmaceutical Sciences, Osaka University, 1-6 Yamadaoka, Suita 565-0871, Osaka, Japan

**Keywords:** chimeric antigen receptor, CAR-T cell therapy, disulphide bonds, glycosylation, hinge domain

## Abstract

To improve the efficacy and safety of chimeric antigen receptor (CAR)-expressing T cell therapeutics through enhanced CAR design, we analysed CAR structural factors that affect CAR-T cell function. We studied the effects of disulphide bonding at cysteine residues and glycosylation in the HD on CAR-T function. We used first-generation CAR[V/28/28/3z] and CAR[V/8a/8a/3z], consisting of a mouse vascular endothelial growth factor receptor 2 (VEGFR2)-specific single-chain variable fragment tandemly linked to CD28- or CD8α-derived HD, transmembrane domain (TMD) and a CD3ζ-derived signal transduction domain (STD). We constructed structural variants by substituting cysteine with alanine and asparagine (putative N-linked glycosylation sites) with aspartate. CAR[V/28/28/3z] and CAR[V/8a/8a/3z] formed homodimers, the former through a single HD cysteine residue and the latter through the more TMD-proximal of the two cysteine residues. The absence of disulphide bonds did not affect membrane CAR expression but reduced antigen-specific cytokine production and cytotoxic activity. CAR[V/28/28/3z] and CAR[V/8a/8a/3z] harboured one N-linked glycosylation site, and CAR[V/8a/8a/3z] underwent considerable O-linked glycosylation at an unknown site. Thus, N-linked glycosylation of CAR[V/28/28/3z] promotes stable membrane CAR expression, while having no effect on the expression or CAR-T cell activity of CAR[V/8a/8a/3z]. Our findings demonstrate that post-translational modifications of the CAR HD influence CAR-T cell activity, establishing a basis for future CAR design.

## 1. Introduction

Chimeric antigen receptor (CAR)-expressing T cell (CAR-T) therapy targeting CD19 has been introduced into clinic settings for the treatment of refractory B cell lymphomas [1,2]. While its novelty and impressive efficacy have thrust CAR-T into the scientific limelight, its serious side effects, including cytokine release syndrome and on-target off-tumour effects, represent a major drawback [3,4]. Consequently, a comprehensive CAR design framework and effective CAR-T cell control methods for minimising side effects are of the utmost necessity. However, information and evidence regarding the relationship between CAR structure and CAR-T cell activity remain scarce. Thus, research into the fine tuning of CAR-T cell activity is lagging behind.

The CAR protein consists of an antigen-recognising domain (ARD), a hinge domain (HD), a transmembrane domain (TMD) and an intracellular signal transduction domain (STD). Single-chain variable fragments (scFvs) and various components from immune-related signalling proteins have been used to construct CARs. We previously used a variety of CAR components to construct several CAR structural variants in an attempt to acquire insight into the relationship between CAR structure and CAR-T cell activity [5,6,7]. However, these efforts were made through activity correlation analyses based on alterations in the primary structure (amino acid sequence) of the CAR protein. To date, analyses of the effects of post-translational modifications on CAR-T cell activity are lacking. Within their original proteins, these borrowed components undergo post-translational modifications, such as disulphide bonding and glycosylation, which are known to play an important role in their function [8,9,10,11,12,13]. It is therefore quite likely that these moieties also undergo post-translational modification as part of the CAR construct, which would in turn affect CAR expression or CAR-T cell function. However, the link between CAR post-translational modification and CAR-T cell activity remains elusive, as CAR design has been largely heuristic in nature.

Thus, in this study, we used first-generation CARs consisting of a mouse vascular endothelial growth factor receptor 2 (VEGFR2)-specific scFv tandemly linked to a CD28- or CD8α-derived HD/TMD and a CD3ζ-derived STD in order to investigate CAR post-translational modifications and their effects on CAR-T cell function.

## 2. Results

### 2.1. Post-Translational Modifications of CAR[V/28/28/3z] and CAR[V/8a/8a/3z]

We sought to investigate the formation of intermolecular disulphide bonds and the occurrence of glycosylation in CAR variants CAR[V/28/28/3z] and CAR[V/8a/8a/3z] constructed in this study. After treating CAR-T protein extracts with deglycosylation enzymes, these were subjected to Western blotting under reducing and non-reducing conditions (Figure 1). Under non-reducing conditions, bands with molecular weights higher than the expected monomeric weights were detected for both CAR[V/28/28/3z] and CAR[V/8a/8a/3z], indicating that both CARs formed complexes via disulphide bonding at cysteine residues. We also treated the protein extracts with PNGaseF, which specifically digests N-linked glycans. In that case, bands of both the monomers and the complex shifted towards the smaller-molecule side of the blot for CAR[V/28/28/3z] as well as CAR[V/8a/8a/3z], indicating that both CARs undergo N-linked glycosylation. Further, after treating protein extracts with Deglycosylation Mix II, which digests both N-linked and O-linked glycans, the same band shift seen with PNGaseF was observed for CAR[V/28/28/3z], revealing that it does not undergo O-linked glycosylation. On the other hand, after treatment of CAR[V/8a/8a/3z] with Deglycosylation Mix II, a downwards band shift of a greater magnitude than that of PNGaseF was observed, indicating that it does undergo O-linked glycosylation. Finally, when glycosylation was removed from both CAR[V/28/28/3z] and CAR[V/8a/8a/3z] under reducing conditions, a band was detected at the molecular weight of the monomer expected from its amino acid sequence. Under non-reducing conditions, a band was observed at almost twice this molecular weight. Taken together, these findings suggest that the CARs are expressed as homodimers at the T cell membrane.

### 2.2. The Effects of Intra-HD Intermolecular Disulphide Bonds at Cysteine Residues on CAR Expression and CAR-T Cell Cytotoxic Activity

We constructed four structural variants of CAR[V/28/28/3z] and CAR[V/8a/8a/3z] by substituting cysteines within their HDs with alanine: CAR[V/28/28/3z](C142A), CAR[V/8a/8a/3z](C178A/193A), CAR[V/8a/8a/3z](C178A) and CAR[V/8a/8a/3z](193A) (Figure 2). The proteins extracted from CAR-T cells expressing these variants were subjected to Western blotting under reducing and non-reducing conditions (Figure 3A). Nearly identical bands were observed for CAR[V/28/28/3z](C142A) under both conditions, suggesting that CAR[V/28/28/3z] forms a homodimer through disulphide bonding at the single cysteine residue within its HD. Similarly, homodimer bands were absent for the CAR[V/8a/8a/3z](C178A/193A) and CAR[V/8a/8a/3z](193A) variants; the former substituted both HD cysteine residues with alanine and the latter substituted only the HD cysteine residue closer to the TMD with alanine. On the other hand, homodimerisation was observed for CAR[V/8a/8a/3z](C178A) in the same manner as for CAR[V/8a/8a/3z]. Thus, it was revealed that only the HD cysteine residue of CAR[V/8a/8a/3z], which is closer to its TMD, contributes to homodimerisation via disulphide bonding.

Next, the expression and antigen-binding ability of these CAR variants were investigated in mouse T cells. All HD-cysteine-substituted variants demonstrated the same expression levels and antigen-binding activity as the original CAR[V/28/28/3z] and CAR[V/8a/8a/3z] proteins (Figure 3B). In addition, the mRNA transcription level from the CAR gene introduced by a retroviral vector (Rv) showed an almost constant value until the sixth day of culture, and no difference was observed between the original CARs and any of the variants (Figure 3C). Substitution of cysteine residues in HD also had no effect on the expression profile over time at the T-cell membrane (Figure 3D).

Antigen-specific activity was analysed using CAR-T cells on the fourth day of culture, as expression levels of various CAR variants were equivalent. Based on BrdU uptake, the proliferative activity of CAR[V/28/28/3z] and CAR[V/8a/8a/3z] was not affected by the lack of cysteine residues (Figure 3E). Both CAR-T cells showed a similar antigen concentration-dependent growth profile. Regarding cytokine production associated with mVEGFR2 stimulation, CAR[V/28/28/3z](C142A)-T cells had significantly lower levels of IFN-γ and IL-2 production than CAR[V/28/28/3z]-T cells (Figure 3F). CAR[V/8a/8a/3z](C178A)-T cells also exhibited lower levels of IFN-γ and IL-2 production than CAR[V/8a/8a/3z]-T cells, while CAR[V/8a/8a/3z](C193A)-T cells demonstrated even lower production than CAR[V/8a/8a/3z](C178A)-T cells. T cells containing the double-substitution variant, (CAR[V/8a/8a/3z](C178A/193A)-T cells) had lower levels of IFN-γ, TNF-α and IL-2 production than CAR[V/8a/8a/3z]-T cells, in addition to lower production of these molecules than T cells with the corresponding single-substituted variants (i.e., CAR[V/8a/8a/3z](C178A)-T cells and CAR[V/8a/8a/3z](C193A)-T cells).

CAR[V/28/28/3z](C142A)-T cells exhibited lower cytotoxic activity than CAR[V/28/28/3z]-T cells (Figure 3G). Further, T cells made to express variants without the ability to homodimerise, i.e., CAR[V/8a/8a/3z](C178A/193A) and CAR[V/8a/8a/3z](C193A), almost completely lost their cytotoxic activity, suggesting that homodimerisation via HD cysteine residues is important for CAR-T cell functional activity. While the CAR[V/8a/8a/3z](C178A) variant is capable of homodimerising, CAR-T cells expressing this variant demonstrated lower cytotoxic activity than those expressing the original CAR[V/8a/8a/3z] variant. We surmise that this difference is related to conformational changes within the HD associated with the amino acid substitution.

Based on these results, it is clear that intermolecular disulphide binding between CAR proteins via HD cysteine residues does not affect CAR protein expression at the T cell membrane. However, this binding was shown to be important for CAR-T cell antigen recognition-associated functional activity.

### 2.3. The Effects of N-Linked Glycosylation at Asparagine Residues in HD on CAR Expression and CAR-T Cell Cytotoxic Activity

We constructed two structural variants of CAR[V/28/28/3z] and CAR[V/8a/8a/3z] by substituting HD asparagines believed to be N-linked glycosylation sites with aspartate: CAR[V/28/28/3z](N130D) and CAR[V/8a/8a/3z](N150D) (Figure 2). The proteins extracted from CAR-T cells expressing these variants were subjected to Western blotting under reducing and non-reducing conditions after deglycosylation treatment (Figure 4A). CAR[V/28/28/3z](N130D) shifted towards the smaller-molecule side relative to CAR[V/28/28/3z], and its band position did not change after treatment with PNGaseF, revealing that CAR[V/28/28/3z] undergoes N-linked glycosylation at the single substituted asparagine residue. Similarly, a downward band shift was observed for CAR[V/8a/8a/3z](N150D) relative to CAR[V/8a/8a/3z], and no change in band position was observed after PNGaseF treatment. Thus, we confirmed that CAR[V/8a/8a/3z] undergoes N-linked glycosylation at the single substituted asparagine residue.

We also carried out Western blot analysis of CAR[V/8a/8a/3z](T154A/S162A), variants of CAR[V/8a/8a/3z] constructed by substitution of an HD threonine and an HD serine with alanine, both believed to be O-linked glycosylation sites. However, these variants formed bands identical to those of the original CAR[V/8a/8a/3z] protein, and treatment with Deglycosylation Mix II caused the bands to shift towards the smaller-molecule side. Thus, we were unfortunately unable to identify O-linked glycosylation sites in CAR[V/8a/8a/3z] (Appendix A).

We investigated the expression and antigen-binding ability of CAR[V/28/28/3z](N130D) and CAR[V/8a/8a/3z](N150D), both of which cannot undergo N-linked glycosylation, in mouse T cells. The expression level of CAR[V/28/28/3z](N130D) at the T cell membrane was lower than that of the original CAR[V/28/28/3z], and its antigen-binding ability was also reduced accordingly (Figure 4B). On the other hand, the expression level and antigen-binding ability of CAR[V/8a/8a/3z](N150D) were both similar to those of the original CAR[V/8a/8a/3z]. In addition, the mRNA expression of the CAR gene introduced by Rv showed an almost constant value until the sixth day of culture, and no difference was observed between the original CAR and each variant (Figure 4C). With respect to their longitudinal expression profiles at the T cell membrane, while CAR[V/28/28/3z](N130D) tended to show a lower level of expression intensity than CAR[V/28/28/3z], the expression level of CAR[V/8a/8a/3z](N150D) was identical to that of CAR[V/8a/8a/3z] (Figure 4D).

Antigen-specific proliferative activity was compared using various CAR-T cells on the fourth day of culture. As a result, CAR[V/28/28/3z](N130D)-T cells showed a slight decrease compared to CAR[V/28/28/3z]-T cells (Figure 4E). This was considered to be due to the difference in the CAR expression level shown in Figure 4D. On the other hand, CAR[V/8a/8a/3z] and CAR[V/8a/8a/3z](N150D) showed similar antigen-concentration-dependent growth profiles. The antigen-stimulation-associated cytokine production of CAR[V/28/28/3z](N130D)-T cells was lower than that of CAR[V/28/28/3z]-T cells only for IFN-γ (Figure 4F). The IL-2 production ability of CAR[V/28/28/3z](N130D)-T cells as well as the IFN-γ and IL-2 production ability of CAR[V/8a/8a/3z](N150D)-T cells were equivalent to those of CAR-T cells expressing the corresponding unsubstituted CAR variants. Further, both CAR[V/28/28/3z](N130D)-T cells and CAR[V/8a/8a/3z](N150D)-T cells exhibited slightly greater TNF-α production than wild-type CAR-T cells. Similarly to the patterns observed for proliferative activity, CAR[V/28/28/3z](N130D)-T cells demonstrated slightly lower cytotoxic activity compared to CAR[V/28/28/3z]-T cells (Figure 4G). In contrast, CAR[V/8a/8a/3z](N150D)-T cells exhibited the same cytotoxic activity as the original CAR[V/8a/8a/3z]-T cells.

In summary, N-linked glycosylation at asparagine residues in the CD28-derived HD promoted CAR expression and CAR-T cell function. In contrast, N-linked glycosylation of the CD8α-derived HD did not greatly affect CAR expression nor CAR-T cell function.

## 3. Discussion

In this study, among the various components of the CAR protein, we focused on the HD in particular. Using two first-generation CARs, CAR[V/28/28/3z] and CAR[V/8a/8a/3z], as our models, we analysed the effects of disulphide bonding at cysteine residues and glycosylation on CAR-T cell activity. As summarised in Figure 5, while homodimerisation of both CAR[V/28/28/3z] and CAR[V/8a/8a/3z] via HD cysteine residues did not affect CAR expression levels or avidity, it clearly contributed to antigen-specific CAR-T cell function. We are not the only group to report that selection of different HD/TMD components can enable the control of CAR-T cell activity without affecting its expression levels [6,14]. CAR is also known to form immune synapses on T cell membranes [15]. Further, CARs with CD28-derived HD/TMD can exert rapid cytotoxic activity by forming such immune synapses more efficiently than CARs with CD8α-derived HD/TMD [16]. Taken together with the current findings, we can surmise that CAR dimerisation via HD disulphide bonding can alter the activity of CAR-T cells without changing their expression levels. This is due to the effect on antigen-recognition-dependent clustering of CARs at the cell membrane and subsequent immune synapse formation.

This study suggested that N-linked glycosylation in CD28-derived HD may contribute to the maintenance of CAR expression on the T cell membrane. We found that the glycosylation-enhanced maintenance of CAR expression affects CAR-T cell function. Herein, Western blotting analysis was performed only immediately after the completion of the gene transfer operation (day 0). It is unclear whether the decrease in CAR expression intensity following the removal of N-linked glycosylation was due to changes in the efficiency of CAR transfer to the cell surface or due to changes in CAR turnover. In the future, we will analyse the intracellular localisation and behaviour of CAR[V/28/28/3z] and CAR[V/28/28/3z](N130D) over time. In this manner, we would like to clarify the role of N-linked glycosylation in the CD28-derived HD. On the other hand, N-linked glycosylation in the HD of CAR[V/8a/8a/3z] had a small effect on the maintenance of CAR expression. Thus, it was not considered to be of importance for CAR-T cell function. The replacement/removal of N-linked glycosylation sites in CD8α-derived HD may represent one approach for improving CAR-T cell homogeneity as a drug formulation. Although it was found that CAR[V/8a/8a/3z] also undergoes O-linked glycosylation, we were unable to identify the amino acid residues at which these modifications occur. Regarding O-linked glycosylation of the original CD8α molecule, it has been reported that glycosylation status changes depending on the T cell activation state and differentiation stage [17]. Although the previous report [18] did not discuss O-linked glycosylation of CAR, it nonetheless demonstrates that even if one O-linked glycosylation site is deleted, another is modified. Therefore, it is possible that the same phenomenon takes place in CAR[V/8a/8a/3z]. In the future, we plan to identify the O-linked glycosylation site in the CD8α-derived HD by using the alanine mutagenesis screening technique or the mass spectrometric approach and to determine its effect on CAR-T cell function. There has been no report on the relationship between the presence or absence of the glycosylation of CAR and CAR-T cell function. Although we have not yet been able to investigate O-linked glycosylation in detail, we think these results are an important stepping stone to presenting its significance, the regulation of glycosylation for optimisation of CAR function and the homogenisation of CAR-T cell quality. In addition, this study was evaluated in a mouse model. The effects of the HD region on CAR-T cells have not been evaluated in either mice and humans, and the results in mice may not be applicable to humans, but we think that the evaluation in humans is also an issue for the future.

It has also been reported that the positional relationship between the CAR extracellular domain and the epitope affects CAR-T cell function [19,20,21]. It is therefore necessary to consider the length of the HD, both in terms of the number of amino acids that comprise it and its folded, three-dimensional structure. It is important to note that the current findings were only obtained from two types of model CARs, both of which have HDs that adopt a random coil structure. In the future, we will prepare more CAR variants with various numbers and combinations of secondary structural units (α-helix and β-strand) in their HDs. In this way, we plan to comprehensively analyse the effects of HD length and flexibility on CAR-T cell activity.

## 4. Materials and Methods

### 4.1. Cell Lines

Human Plat-E cells were obtained from Cell Biolabs (San Diego, CA, USA) and cultured in Dulbecco’s modified Eagle’s medium supplemented with 10% foetal bovine serum (FBS, Thermo Fisher Scientific, Waltham, MA, USA), 1 µg/mL puromycin (Merck, Darmstadt, Germany) and 10 µg/mL blasticidin (FUJIFILM Wako Pure Chemical, Osaka, Japan). Murine EL4 cells were obtained from the Cell Resource Center for Biomedical Research, Institute of Development, Aging, and Cancer, Tohoku University (Sendai, Japan). These were cultured in complete RPMI 1640 (cRPMI) medium (FUJIFILM Wako Pure Chemical, Osaka, Japan) supplemented with 10% FBS, Antibiotic-Antimycotic Mixed Stock Solution (100×) 5 mL (10,000 U/mL penicillin, 10,000 µg/mL streptomycin, 25 µg/mL amphotericin B, Nacalai Tesque, Kyoto, Japan) and 50 µM 2-mercaptoethanol (2-ME, Merck, Darmstadt, Germany). Murine-VEGFR2-expressing EL4 (VEGFR2^+^ EL4) cells were generated via Rv transduction (containing the VEGFR2 gene and a puromycin resistance cassette) and were grown in cRPMI medium, supplemented with 5 µg/mL puromycin. All cells were maintained in a humidified atmosphere of 5% CO_2_ at 37 °C.

### 4.2. Mice

Female C57BL/6 mice (6 weeks old) were purchased from SLC (Hamamatsu, Japan) and were maintained in the experimental animal facility at Osaka University.

### 4.3. Construction of CAR Structural Variants

The basic structure of CAR followed the previous report [7]. The notation and structure of various structurally modified CARs and the amino acid sequence of the HD part are summarised in Figure 2. An HA-tag was incorporated into the N-terminus of CAR. We took two CARs consisting of a mouse-VEGFR-specific scFv tandemly linked to a CD28- or CD8α-derived HD/TMD and a CD3ζ-derived STD as our basic structural units. We then constructed six structural variants of these CARs by substituting cysteines in the HD with alanine and by substituting various amino acids in a region described as a potential glycosylation site in the literature [13] as well as in protein databases. In the CD28-derived HD, we substituted one cysteine residue with one alanine and one asparagine that is an N-linked glycosylation site with aspartate. In the CD8α-derived HD, we substituted either one, the other, or both cysteine residues with alanine, in addition to substituting one asparagine, an N-linked glycosylation site, with aspartate.

### 4.4. Production of CAR-T Cells

Murine CAR-T cells were produced as previously described [22]. Briefly, a murine leukaemia virus-derived pMXs-Puro Retroviral Vector (Cell Biolabs, San Diego, CA, USA) was used as a plasmid for Rv production. The Rv packaging CAR gene was produced by transfecting Plat-E cells with the pMXs-Puro/CAR. The culture supernatant of Plat-E cells obtained 48 h later was filtered through a 0.45 μm filter and used as an Rv solution for gene transfer. Murine T cells were activated by incubation with an anti-CD3ε mAb (clone 145-2C11, Bioxcell, West Lebanon, NH, USA) and anti-CD28 mAb (clone 37.51, Bioxcell, West Lebanon, NH, USA) and then transduced with Rv-bound Retronectin (Takara Bio, Kusatsu, Japan) under anti-CD3ε/CD28 mAbs stimulation. The gene-transduced cells were cultured in cRPMI medium, supplemented with 5 μg/mL puromycin. We defined the end of the 24 h culture as the end of the gene transfer operation on the 0th day (day 0).

### 4.5. Reverse Transcription and Quantitative Polymerase Chain Reaction

mRNA was extracted from genetically modified T cells using TRIzol reagent (Thermo Fisher Scientific, Waltham, MA, USA). Complementary DNA (cDNA) was prepared through a reverse transcription reaction using Super Script III Reverse Transcriptase (Thermo Fisher Scientific, Waltham, MA, USA) after DNase treatment. The CAR gene-encoded mRNA was detected via the TaqMan Gene Expression Assay (Thermo Fisher Scientific, Waltham, MA, USA) with primers and probes that bind to the mouse-VEGFR2-specific scFv. We also detected mRNA encoded by the *GAPDH* gene as an endogenous control. The expression level of each gene was measured on a CFX96 Real-Time PCR Detection System (Bio-Rad Laboratories, Hercules, CA, USA).

### 4.6. Flow Cytometry (FCM) Analysis for Surface Expression and Antigen-Binding Capacity of CAR

Using a staining buffer (PBS supplemented with 2% FBS and 0.05% NaN_3_), 5 × 10^5^ T cells were first treated with a TruStain fcX (anti-mouse CD16/32) Antibody (clone 93; BioLegend, San Diego, CA, USA). Staining was then performed using the Zombie Aqua Fixable Viability Kit (BioLegend, San Diego, CA, USA), an eFluor450-conjugated anti-mouse CD4 mAb (clone GK1.5; Thermo Fisher Scientific, Waltham, MA, USA) and a PE-Cy7-conjugated anti-mouse CD8 mAb (clone 53-6.7; Thermo Fisher Scientific, Waltham, MA, USA). Since we incorporated an HA-tag at the CAR N-terminus, we used APC-conjugated anti-HA mAb (clone GG8-1F3.3.1, Miltenyi Biotec, Bergisch Gladbach, Germany) and APC-conjugated mouse IgG1κ Isotype Control (clone P3.6.2.8.1, Thermo Fisher Scientific, Waltham, MA, USA) to detect CAR expression. The geometric mean fluorescence intensity (GMFI) of live cells, lymphocytes and CD8-positive fractions was measured via FCM analysis using anti-HA-Tag antibody or isotype control. The GMFI ratio was calculated from the following formula, and this value was quantified as the CAR expression intensity.
[GMFI ratio = GMFI when using anti-HA-Tag antibody/GMFI when using isotype control]

To analyse the avidity of our CARs for mouse VEGFR2, we used a staining buffer (PBS supplemented with 2% FBS and 0.05% NaN_3_) and first treated 5 × 10^5^ T cells with TruStain fcX (anti-mouse CD16/32) Antibody (clone 93; BioLegend, San Diego, CA, USA). Then, the CAR-antigen reaction was carried out using recombinant mouse VEGFR2-Fc Chimera (BioLegend, San Diego, CA, USA). As a control, we subjected a group of cells to the same procedures without addition of the antigen. Excess antigen was removed by washing with staining buffer, and staining was performed by adding Zombie Aqua Fixable Viability Kit, eFlu-or450-conjugated anti-mouse CD4 mAb and PE-Cy7-conjugated anti-mouse CD8 mAb. The recombinant mouse VEGFR2-Fc chimera we used contained a His-tag. Therefore, we used Alexa Fluor-conjugated anti-His-tag mAb (clone OGHis, MBL) to detect antigens bound to the CAR. The GMFI of live cells, lymphocytes and CD8-positive fractions with or without antigen was measured via FCM analysis. The GMFI ratio was calculated via the following formula and quantified as the antigen-binding capacity of the CAR:[GMFI ratio = GMFI when antigen is added/GMFI when antigen is not added]

BD FACS Canto II (BD Biosciences, Franklin Lakes, NJ, USA) was used for FCM analysis.

### 4.7. Protein Extraction from CAR-T Cells and Enzymatic Deglycosylation

Total protein was extracted from CAR-T cells immediately after the gene transfer operation (day 0) using cell lysis buffer (50 mM Tris-HCl (pH 7.4) supplemented with 150 mM NaCl, 1 mM EDTA, 5% glycerol, 1% Triton-X). Thereafter, protein levels were quantified via the Lowry method using a DC Protein Assay kit (Bio-Rad Laboratories, Hercules, CA, USA).

A sample from which N-linked glycans had been removed was prepared by treating the protein extract sample from CAR-T cells with PNGaseF (New England Biolabs, Ipswich, MA, USA). In addition, a sample from which both N-linked and O-linked glycans had been removed was prepared via treatment with Deglycosylation Mix II (New England Biolabs, Ipswich, MA, USA).

### 4.8. Western Blotting

Lysate sample was mixed with Laemmli sample buffer (Bio-Rad Laboratories, Hercules, CA, USA) containing 20% 2-mercaptoethanol or not, and heat-denatured. Next, lysate samples were separated by sodium dodecyl sulphate–polyacrylamide gel electrophoresis with 7.5% SuperSep Ace (FUJIFILM Wako Pure Chemical, Osaka, Japan) and transferred to a polyvinylidene difluoride membrane (GE Healthcare, Menlo Park, CA, USA) using a Trans-Blot SD Cell (Bio-Rad Laboratories, Hercules, CA, USA). The membrane was blocked in 4% Block Ace (KAC Co., Kyoto, Japan), and then reacted with HA-Tag Rabbit mAb (clone C29F4, Cell Signaling Technology, Danvers, MA, USA) in tris-buffered saline containing 0.05% Tween-20 and 1% bovine serum albumin. The membrane was then washed and reacted with anti-rabbit IgG HRP-linked antibody (Cell Signaling Technology, Danvers, MA, USA). Immunoreactive proteins on the membrane were detected using ImmunoStar Zeta (FUJIFILM Wako Pure Chemical, Osaka, Japan) and ImageQuant LAS4010 (GE Healthcare, Menlo Park, CA, USA).

### 4.9. BrdU Proliferation Assay and Cytokine-Enzyme-Linked Immunosorbent Assay (ELISA)

Four days after Rv transduction, CAR-T cells were cultured for 24 h on a plate coated with VEGFR2-Fc (2–2000 ng/mL). Proliferation activity was measured via BrdU uptake ELISA (Sigma-Aldrich, St. Louis, MO, USA). IFN-γ, TNF-α and IL-2 levels in the supernatants were determined using the OptiEIA™ ELISA Set (BD Biosciences, Franklin Lakes, NJ, USA).

### 4.10. Cytotoxicity Assay

EL4 cells were stained with Violet Proliferation Dye 450 (BD Biosciences, Franklin Lakes, NJ, USA), and EL4 cells expressing mVEGFR2 were stained with Cell Proliferation Dye eFluor 670 (Thermo Fisher Scientific, Waltham, MA, USA). These target cells were co-cultured in batches of 1 × 10^5^ cells for 18 h with enough CAR-T cells to match the effector/target ratio of each well. Four days after the end of the gene transfer operation, CAR-T cells were used for this experiment. Count Bright Absolute Counting Beads (Thermo Fisher Scientific, Waltham, MA, USA) were added to unify the analysis volume of all samples, and 7-AAD Viability Staining Solution (BioLegend, San Diego, CA, USA) was added to stain dead cells. FCM analysis was then performed and terminated when 1000 Count Bright Absolute Counting Beads were detected in each sample. The ratio (R) of the number of mVEGFR2-expressing EL4 cells to the number of living EL4 cells was calculated for each well, and the cytotoxic activity was calculated from the following formula.
Cytotoxicity (% of lysis) = R (control well) − R (test well)/R (control well) × 100
control well: seeded only with target cells; test well: seeded with both effector and target cells.

### 4.11. Statistical Analysis

All experimental data are presented as the mean ± SD. Statistical significance was evaluated via Welch’s *t*-test or Tukey’s test.

## 5. Conclusions

The present study revealed that post-translational modification within the hinge domain of the CAR protein affects CAR-T cell activity. In the future, we expect that tighter control of CAR post-translational modifications will be conducive to homogenising CAR-T cell preparations and minimising the side effects of CAR-T cell therapy.

## Figures and Tables

**Figure 1 ijms-23-04056-f001:**
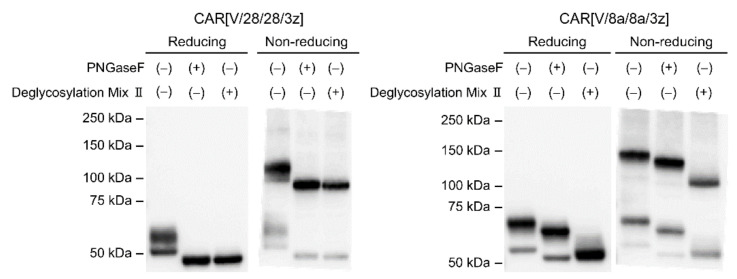
Analysis of cysteine-mediated disulphide bonds and glycosylation in the HD of CAR[V/28/28/3z] and CAR[V/8a/8a/3z]. SDS-PAGE and Western blotting analysis showing the expression of two CARs in whole CAR-T cell lysate on day 0 (24 h after transduction).

**Figure 2 ijms-23-04056-f002:**
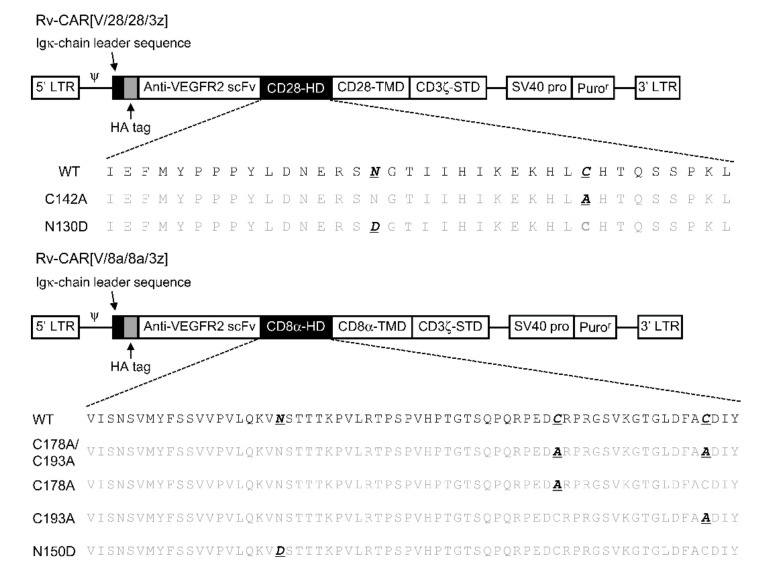
Illustration of the retroviral vector construct containing the gene encoding a VEGFR2-specific CAR and the amino acid sequence of the HD.

**Figure 3 ijms-23-04056-f003:**
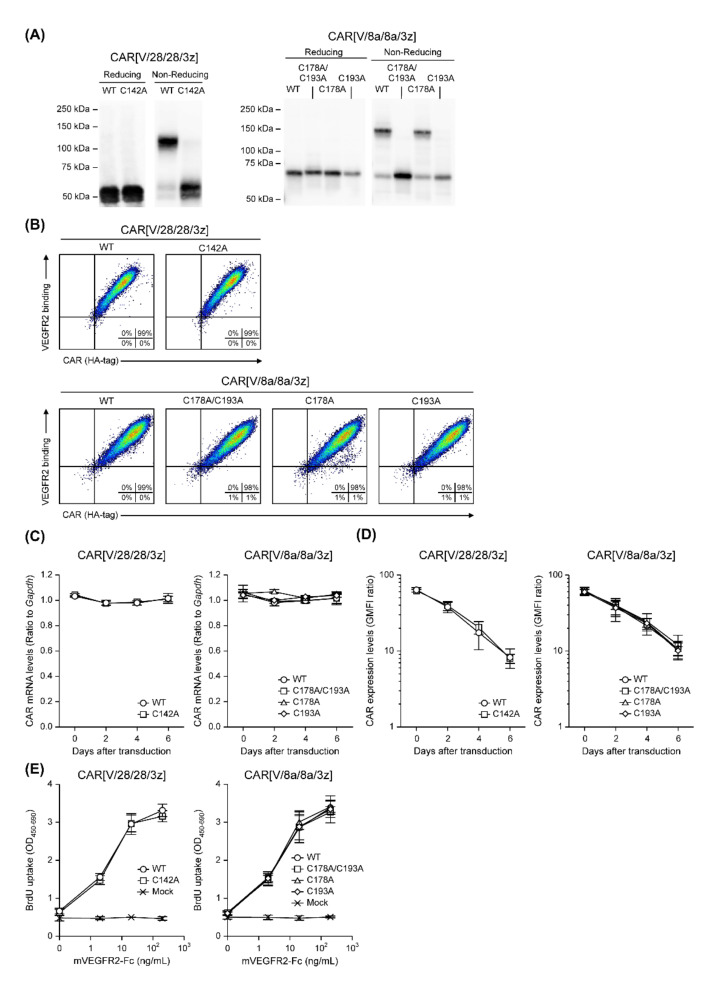
Influence on expression and antigen-specific function of CAR[V/28/28/3z] or CAR[V/8a/8a/3z] following deletion of the cysteine in HD. (**A**) SDS-PAGE and Western blotting analysis showing the expression modality of six CAR variants in whole CAR-T cell lysate on day 0. (**B**) The surface expression and antigen-binding ability were determined via FCM analysis on day 0. All cells were pre-gated with live cells, lymphocytes, and CD8α^+^ cell. (**C**) CAR mRNA expression was analysed via RT-qPCR, and their transcription levels were calculated relative to GAPDH mRNA as an endogenous control. (**D**) CAR expression on T cells was analysed via FCM. Each CAR expression level was calculated from the ratio of geometric mean fluorescence intensity (GMFI) when stained with the anti-HA-tag mAb to GMFI when stained with the isotype control antibody. (**E**) Proliferation activity of HD-modified CAR-T cells following mVEGFR2 stimulation. (**F**) Cytokine production ability of the above cells following mVEGFR2 stimulation. (**G**) Cytotoxic activity of the above cells four days after Rv transduction against mVEGFR2^+^ EL4 cells. The data were obtained from three independent tests. Statistical analysis was performed using Welch’s *t*-test (* *p* < 0.05 and ** *p* < 0.01 versus WT) in CAR[V/28/28/3z] or Tukey’s test (* *p* < 0.05 and ** *p* < 0.01 versus WT, ^†^
*p* < 0.05 and ^††^
*p* < 0.01 versus C178A/C193A, ^‡‡^
*p* < 0.01 versus C193A) in CAR[V/8a/8a/3z].

**Figure 4 ijms-23-04056-f004:**
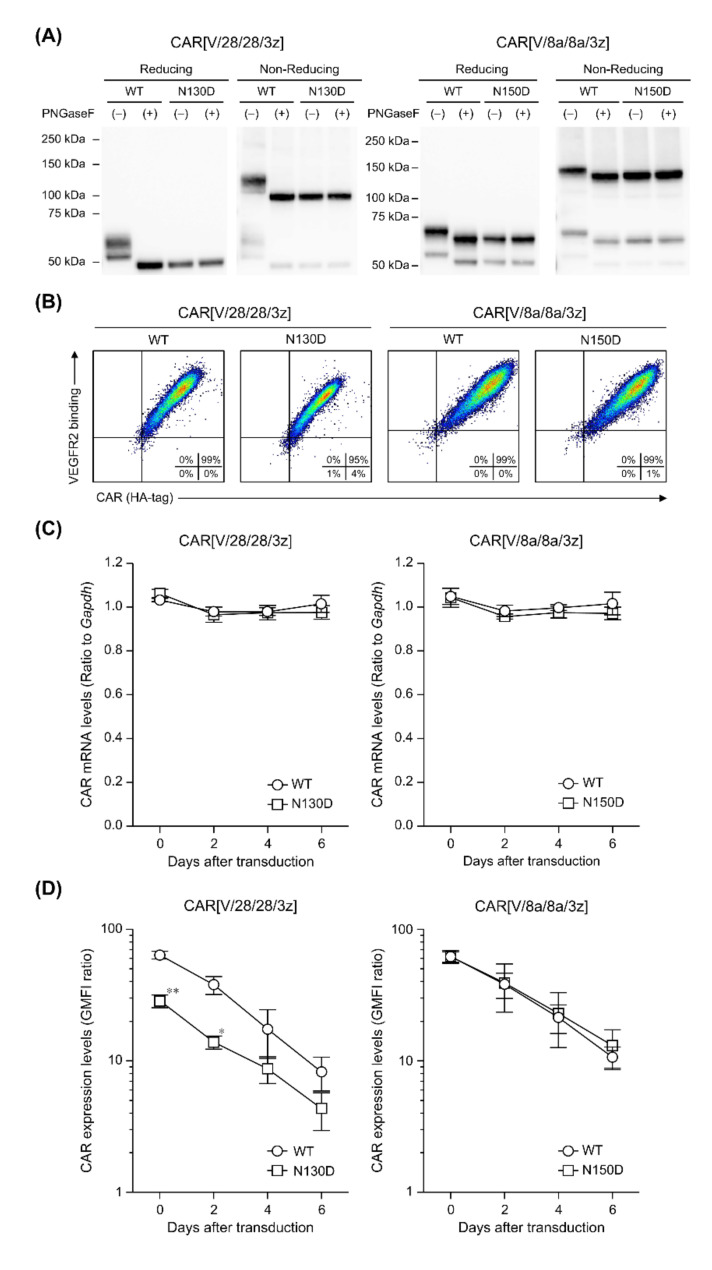
Influence on the expression and antigen-specific function of CAR[V/28/28/3z] or CAR[V/8a/8a/3z] after deletion of N-linked glycosylation in HD. (**A**) SDS-PAGE and Western blotting analysis showing the expression modality of four CAR variants in the whole CAR-T cell lysate on day 0. (**B**) The surface expression and antigen-binding ability were determined via FCM analysis on day 0. All cells were pre-gated with live cells, lymphocytes, and CD8α^+^ cells. (**C**) CAR mRNA expression was analysed via RT-qPCR and normalised relative to GAPDH mRNA as an endogenous control. (**D**) CAR expression on T cells was analysed via FCM. CAR expression level was calculated from the ratio of GMFI when stained with the anti-HA-tag mAb to GMFI when stained with the isotype control antibody. (**E**) Proliferation activity of HD-modified CAR-T cells following mVEGFR2 stimulation. (**F**) Cytokine production ability of the above cells following mVEGFR2 stimulation. (**G**) Cytotoxic activity of the above cells four days after Rv transduction against mVEGFR2^+^ EL4 cells. The data were obtained from three independent tests. Statistical analysis was performed using Welch’s *t*-test (* *p* < 0.05 and ** *p* < 0.01 versus WT).

**Figure 5 ijms-23-04056-f005:**
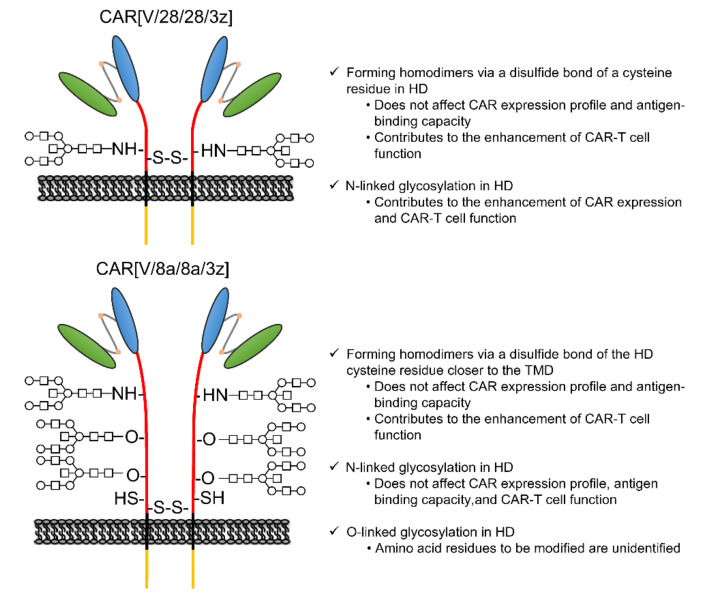
Role of disulphide bonding via cysteine residue and glycosylation in the HD on CAR[V/28/28/3z] and CAR[V/8a/8a/3z] expression and function.

## Data Availability

Not applicable.

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
