# Peer review of "The Effects of Chimeric Antigen Receptor (CAR) Hinge Domain Post-Translational Modifications on CAR-T Cell Activity"

_ijms, 2022, doi:10.3390/ijms23074056_

Round 1
Reviewer 1 Report
The authors in the manuscript entitled “The effects of chimeric antigen receptor (CAR) hinge domain post-translational modifications on CAR-T cell activity” decipher the importance of two post translational modifications in the expression and functional activity of two first-generation model CARs. Based on their biochemical and cell cellular assays, the disulfide bond seems less critical for expression of both the CARs but seems critical for functional signaling activity. The N-linked glycosylation of CD28HD CAR seems to be critical both for the expression and functional activity. The N-linked glycosylation of CD8HD CAR, however, does not seem to be critical both for the expression and functional activity. The author proposed that the O linked glycosylation may be critical for its activity. Since the premise of the manuscript is based on “We studied the effects of disulfide bonding at cysteine residues and glycosylation in the HD on CAR-T function.”, I think it is important to understand the impact of O-linked glycosylation, if any, in the expression and functional activity of the CD8HD CAR. A full Ala scan is a good idea but can also use other approaches such as mass spectrometry. The issue whether O-linked glycosylation is truly affecting the activity should be clarified in this manuscript i.e., identification of the O-glycosylation site(s).
Is it possible that when both the N-linked and the O-linked glycosylation sites are present in CAR HDs, then O-linked glycosylation is predominantly critical for expression and activity? Could the authors artificially introduce an O-linked glycosylation site on the CD28HD CAR and see its effect?
It would be nice if the authors show Figure S2 as Fig 5 in the main text. This is a nice summary of the work. The readers would really benefit having it as a final figure.
The discussion section can be elaborated in light with the findings from other research groups. How the research findings compare with previously published literature.
Overall, the manuscript is well written, and the work is well described. It needs a few additional experimental support.
Reviewer 2 Report
The authors study the effect of two types of pos-translational modifications on the function of two CAR-T cell receptor constructs. A cystein disulfide bond and glycosylation sites. They show the existence of these modifications using western blots (that are high quality and well explained), and then show the in vitro efficacy of these CAR T-cell constructs with or without removal of these post-translational modifications using point-mutations. The authors evaluate the expression of the modified CAR-T cell receptors, the capacity of the CAR T-cells to proliferate upon stimulation and their capacity to produce cytokines. Altogether, they could show that these modifications do not impact the CAR T-cell expression but rather impact the construct's function to activate T cells (with more effect on the disulfide bond and different levels of impact on the different glycosylation sites). It is of great therapeutical interest to determine candidate positions to be improved in the design of CAR T cell constructs, and therefore this manuscript is of interest for the community.
Altogether, the manuscript is well written, the authors use the proper methods to evaluate the function of modified CAR constructs, and the results are clear and support the claims. This was a lot of work.
Only one minor point, Figure S2 is a nice summary and could benefit to be moved to the main text in the conclusion.
No additional experiment seems needed.
Author Response
We are pleased to note your favorable comments. We have responded to your comment below.
Only one minor point, Figure S2 is a nice summary and could benefit to be moved to the main text in the conclusion.
Thank you for your suggestion. Figure S2 has been added to the main text as Figure 5.
Reviewer 3 Report
In the manuscript entitled “The effects of chimeric antigen receptor (CAR) hinge domain post-translational modifications on CAR-T cell activity” Hirobe S and colleagues evaluated the impact of post-translational modifications in the hinge region of mouse vascular endothelial growth factor receptor 2 (VEGFR2)-CAR on CAR-T cell functions, aimed to provide evidence that CAR T cells activity may be influenced by CAR structure and therefore improved by rational optimization in CAR design.
In the study, the authors utilized two basic VEGFR2-CAR structures composed of mouse VEGFR2 single-chain linked to CD3z endodomain through a hinge (HD) and transmembrane (TM) moieties derived from CD28 or CD8a. The authors designed six structural variants by substituting crucial amino acid residues for disulphide bonding and glycosylation in the CD8a or CD28 hinge domain of the two VEGFR2-CAR basic structures.
In a side-by-side analysis, Hirobe S et al. compared the CAR-V/28/28/3z and CAR-V/8a/8a/3z variants with their respective controls and defined the effect of the amino acid replacements on i) homodimer formation; ii) expression on mouse T cells and antigen-binding capacity; ii) CAR-T cell proliferation; iii) cytokine release, and iv) cytotoxic activity of VEGFR2-CAR-T cells against target cells.
The manuscript is original, well organized, and scientifically relevant. The abstract is clearly written, the title is appropriate to the text content and the cited references are current. The rationale is sound and the experimental plan is timely.
The authors have recently published two similar papers showing the impact of the primary structure of CAR hinge, transmembrane (Fujiwara K et al. Cells, 2020) or signal transduction domain (Fujiwara K et al. Int. J. of Mol. Sci, 2021) on CAR-T cell functions. However, the analysis of the effect of post-translational modifications described in the submitted manuscript is interesting and implements the information provided by the previous papers.
The figures are intelligible and clearly described.
Overall, the paper is clear and the contents are well-explained, however some concerns are necessary.
Major concerns:
1) The study is performed utilizing murine VEGFR2-CAR expressed in murine T cells, and the mutated amino acid residues involved in homodimerization and glycosylation refer to the murine protein sequences. Could the information raised by the results in a mouse model be transferred to a human CAR? Do you expect to replicate the results in a human CAR-T cell model? Is there any information in the literature regarding the relationship between post-translational modifications in human CAR structure and CAR-T cell activity? Adding some sentences with appropriate references in the Discussion section could improve the relevance of the study.
2) Figure 3, panel (A). The western blot image showing the expression of CAR V/28/28/3z (WT and with mutation C142A) in reducing condition and the corresponding image provided in the original form differ (see the file containing original western blotting images).
Minor concern:
Page 11, section 4.3 “Construction of CAR….”. The description of the construction of the two basic CAR structures is not enough accurate. Add details (e.g. origin of CAR fragments, sequence references, cloning strategy..)

Round 2
Reviewer 1 Report
I am satisfied with the responses and corrections from the authors.